# A sensitive mNeonGreen reporter system to measure transcriptional dynamics in *Drosophila* development

Stefano Ceolin[1], Monika Hanf[1], Marta Bozek [1], Andrea Ennio Storti[1], Nicolas Gompel [2], Ulrich Unnerstall[1], Christophe Jung [1✉] & Ulrike Gaul[1,3]

The gene regulatory network governing anterior–posterior axis formation in *Drosophila* is a well-established paradigm to study transcription in developmental biology. The rapid temporal dynamics of gene expression during early stages of development, however, are difficult to track with standard techniques. We optimized the bright and fast-maturing fluorescent protein mNeonGreen as a real-time, quantitative reporter of enhancer expression. We derive enhancer activity from the reporter fluorescence dynamics with high spatial and temporal resolution, using a robust reconstruction algorithm. By comparing our results with data obtained with the established MS2-MCP system, we demonstrate the higher detection sensitivity of our reporter. We used the reporter to quantify the activity of variants of a simple synthetic enhancer, and observe increased activity upon reduction of enhancer–promoter distance or addition of binding sites for the pioneer transcription factor Zelda. Our reporter system constitutes a powerful tool to study spatio-temporal gene expression dynamics in live embryos.

[1] Gene Center and Department of Biochemistry, Center for Protein Science Munich (CIPSM), Ludwig-Maximilians-Universität München, Feodor-Lynen-Strasse 25, 81377 München, Germany. [2] Faculty of Biology, Ludwig-Maximilians-Universität München, Großhaderner Str. 2, 82152 Planegg-Martinsried, Germany. [3] Deceased: Ulrike Gaul. ✉email: jung@genzentrum.lmu.de

The precise spatio-temporal control of gene expression by transcriptional enhancers is key to a variety of biological processes, from animal development to cancer biology[1,2]. A striking example is the organization of the anterior-posterior body axis in the early stages of *Drosophila melanogaster* (*D. melanogaster*) embryonic development. Here, a hierarchically organized network of transcription factors (TF) rapidly subdivides the embryo into progressively smaller regions of expression, which prefigure the segmental body plan of the larva[3,4]. The process is mediated by a large number of transcriptional enhancers, which typically receive input from multiple TF to drive the expression of single elements or portions of complex downstream gene expression patterns with stunning precision[5]. This so-called segmentation network has long served as a key paradigm to decipher how transcriptional control is mediated by enhancers[6,7]. While much progress has been made, culminating in computational models that predict the expression of segmentation enhancers from their sequence with reasonable accuracy[8,9], we are still far from an exact quantitative understanding of how expression is "computed" through the combinatorial occupancy of TF molecules at often long stretches of enhancer DNA, and their interplay with nucleosomes and other features of chromatin organization. To achieve further progress, it will be vital to systematically investigate the effects of modulations of all relevant features of enhancer sequences, such as number, strength, and arrangement of TF binding sites in either mutated native or synthetic enhancer sequences[10]. This in turn requires methods to efficiently quantify the transcriptional activity induced by enhancers at high spatial and temporal resolution, and with medium to high throughput.

Over the years, techniques to measure transcriptional activity have steadily improved. Historically, in situ hybridization in fixed embryos has been the method of choice for the determination of mRNA expression patterns. More recently, single molecule fluorescent in situ hybridization has reached single molecule sensitivity, measuring both cytoplasmic and nascent mRNA transcripts[11]. However, as sensitive as these labeling methods can be, they all rely on the staining of fixed embryos. Thus, the measurement of expression at multiple time points is labor-intensive and subject to difficulties in precise staging, which limits their usefulness in studying transcriptional dynamics. Using live markers is the obvious solution, and fluorescent protein reporters such as GFP have been used extensively to study gene expression dynamics. However, their application in early *D. melanogaster* embryos has been hampered by the fact that the maturation time of most fluorescent proteins is slow relative to the very fast dynamics of gene/enhancer expression, leading to delayed and unfaithful representations of patterns[12]. To overcome this problem, researchers have turned to in vivo mRNA labeling. The MS2-MCP system[13] is a technique that directly captures the temporal dynamics of an enhancer's activity in living cells and organisms. It is based on the coexpression of two components: an MCP-GFP fusion protein, which is provided maternally, and a series of multiple RNA hairpins (MS2 loops) integrated into the sequence of the transgene of interest. As they form, the hairpins are recognized and bound by MCP-GFP; the resulting local accumulation of MCP-GFP molecules is detected as fluorescent spots over a background of the unbound MCP-GFP. This inherent fluorescence background limits the sensitivity of detection. While in cell culture single-molecule sensitivity can be reached, the detection threshold in *D. melanogaster* embryos is much higher, due to more difficult imaging conditions (e.g. light scattering, higher autofluorescence background). Despite this limitation[14], the MS2-MCP system has allowed to unravel new aspects of enhancer activity[13,15] in *D. melanogaster* embryos, such as the role of enhancers in controlling transcriptional bursting[16].

Here, we present a reporter system for tracking enhancer expression that aims to be both more sensitive and better suited to large scale investigations of enhancer function than the MS2-MCP system. Our approach is based on expression of an optimized reporter protein that consists of the bright and fast maturing fluorescent protein mNeonGreen[17], coupled to nuclear localization signals (NLSs) and sequences boosting translational efficiency. Enhancer-driven expression of the reporter is monitored by confocal fluorescence microscopy. Since protein concentration is only an indirect readout of transcription, we reconstruct mRNA levels by analyzing the time course of mNeonGreen fluorescence intensity with a model of ordinary differential equations (ODEs). We validate our approach by comparing our data with those obtained with the MS2-MCP system for the well-studied *hunchback anterior* (*hb_ant*) enhancer[8], and by measuring the activity of the *Kr_CD2* enhancer. In addition, we challenge the sensitivity of our reporter system by measuring the weak expression pattern driven by a short synthetic enhancer carrying three binding sites for the maternal activator Bicoid (Bcd). Finally, we use our reporter to quantify how structural or compositional changes affect the activity of this simple enhancer. We measure quantitative differences in the activity of two variants, one where the enhancer-promoter distance is reduced and another in which three binding sites for Zelda (Zld), thought to promote an open chromatin structure[18], are added. Our data show that the distance between an enhancer and its target promoter affects expression strength, and that Zld influences both the intensity and dynamics of Bcd-dependent transcription.

## Results

**A mNeonGreen reporter system to measure enhancer activity in live *Drosophila melanogaster* embryos.** Tracking the rapid spatiotemporal dynamics of enhancer activity during the early development of *D. melanogaster* embryos requires a quantitative, sensitive, and scalable method. We reasoned that a good starting point might be found among the faster-maturing fluorescent proteins that have been developed in recent years. We considered various fast-maturing fluorescent proteins and settled on the recently discovered fluorescent protein mNeonGreen[17], which has the highest known ratio of molecular brightness over maturation time[19] and has already been used successfully as a fluorescent tag and expression reporter[20]. To optimize the reporter for our purposes, we added a number of modifications: In order to minimize the diffusion of the reporter protein away far from the region where it is synthesized, which can occur due to the syncytial nature of the blastoderm embryo, we added multiple NLSs at both the C- and N-terminus (Fig. 1a). The enrichment of the reporter protein in the nucleus has the additional advantage of increasing the signal intensity over the embryo's autofluorescence background. In preliminary experiments (see Supplementary Information), we tested different configurations of NLS of various classes[21] in *D. melanogaster* S2 cells and selected the one achieving the highest ratio of nuclear to cytoplasmic mNeonGreen fluorescence, namely two NLS sequences downstream and one upstream of the mNeonGreen coding sequence (Fig. 1a and "Methods"). Second, to boost the signal produced by enhancer activation, we codon-optimized the coding sequence ("Methods"), and included a translational enhancer at the 5′UTR (Fig. 1a), which has been proven to increase transgene translation in *D. melanogaster*[22]. Finally, we coupled the construct to a strong synthetic promoter (DSCP)[23]. We refer to the full sequence of the reporter construct including 3′UTR and 5′UTR as *mNeonRep*.

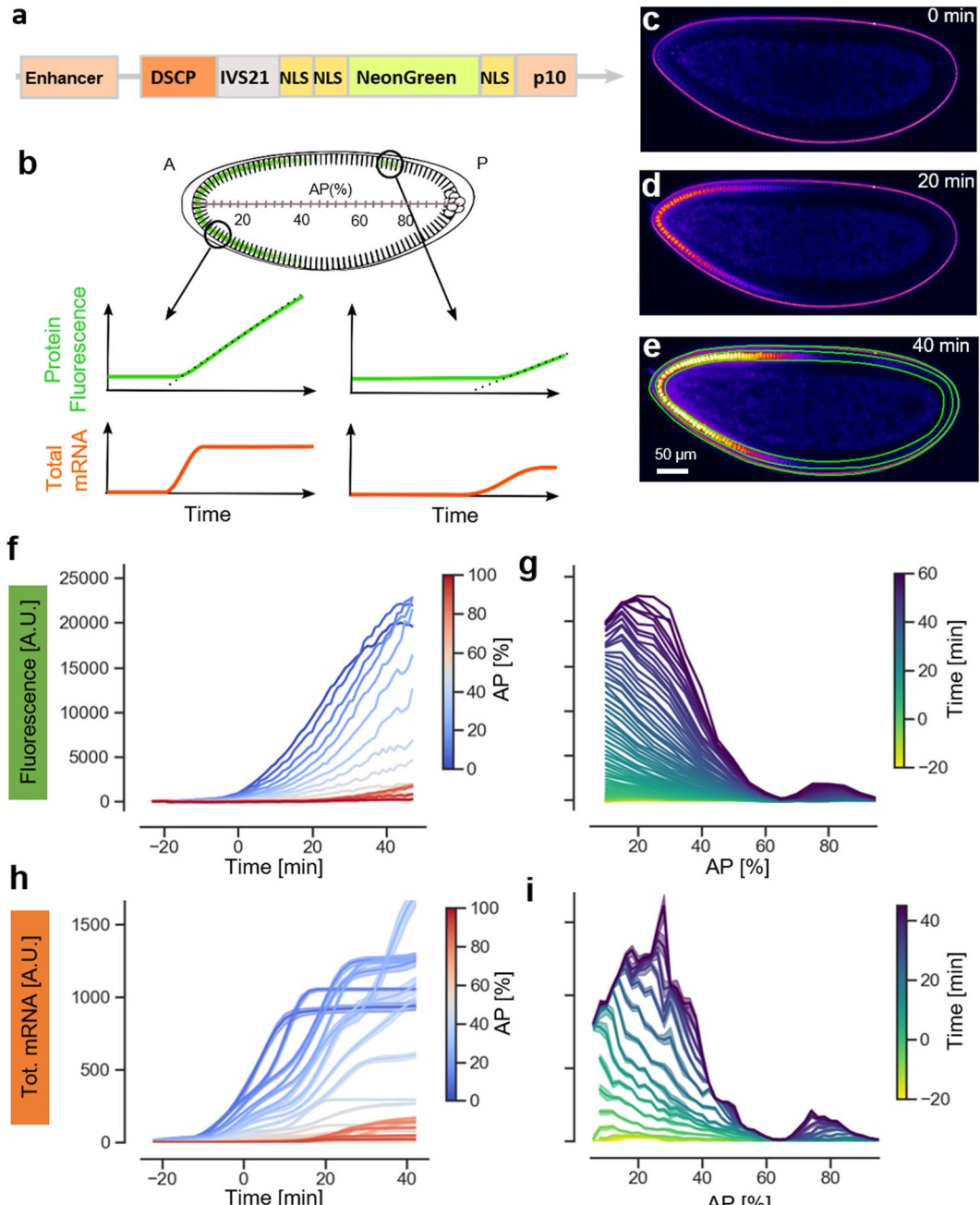

**Fig. 1 A mNeonGreen reporter system to measure enhancer expression in living Drosophila melanogaster embryos. a** Composition of the mNeonGreen reporter construct including: Enhancer, basal DSCP promoter, the translation enhancer sequence IVS21 at the 5′UTR, the mNeonGreen fused to three nuclear localization signals and the terminator sequence p10. **b** Illustration of the use of a fluorescent protein as a transcriptional reporter. The time course of protein fluorescence (in green) is different in different portions of the embryo and carries information on the underlying dynamics of the total mRNA production (in orange), which can be computed using the reconstruction algorithm described in Supplementary Fig. 1. **c–e** Representative confocal slices of embryos carrying hb_ant-mNeonRep at three different time points during embryo development, showing mNeonGreen fluorescence in false colors. **f** Fluorescence time course traces corresponding to the average signal in 2% bins along the AP axis of the embryo, color coded by their position along the axis. **g** Fluorescence expression patterns along the AP axis. Each track corresponds to a different time of embryo development. **h** Time course of cumulative mRNA production in 2% bins along the AP axis of the embryo, color coded by their position along the axis. **i** Total mRNA production patterns at selected times of embryo development.

A protein reporter does not provide a direct readout of transcription, however mRNA concentrations can be reconstructed from the time course of protein concentrations, if relevant kinetic parameters are known (Fig. 1b). To do so, we adapted an approach used to reconstruct promoter activation and deactivation dynamics from protein expression data in bacterial cell cultures[24]. We modeled expression and maturation of the mNeonGreen reporter with a set of linear ODE (Supplementary

Fig. 1 and "Methods"). The model explicitly takes into account the rate of protein maturation and the degradation rates of both protein and mRNA. We measured these parameters in a dedicated set of experiments, in which alpha-amanitin, which blocks transcription, or cycloheximide, which blocks translation, were injected into embryos (Supplementary Fig. 2). We find a maturation time for the mNeonGreen protein of about 7 min, which is in line with previous estimates. Using linear inversion

and a regularized non-negative least square algorithm, the ODE model then allows us to derive the mRNA concentration, the rate of instantaneous mRNA production, as well as the cumulative or total mRNA production from the reporter protein fluorescence for a given time window. To verify the robustness of this data analysis pipeline, we implemented a bootstrapping algorithm (Supplementary Fig. 1 and "Methods") that estimates a confidence interval for the inferred mRNA levels.

We tested the performance and reliability of this reconstruction algorithm with different sets of simulated data and evaluated the impact of the different levels of signal-to-noise ratio (SNR) that we typically observe with our reporter. In addition, we tested the ability of the algorithm to discriminate between different dynamics of enhancer activity, and in particular, to detect when an enhancer is turning on or off (Supplementary Figs. 3–5). Based on these analyses, we estimate that for an SNR like the one obtained for the *hb_ant* enhancer, bursts of changes in transcription can be reliably reconstructed at a resolution of about 7 min (Supplementary Fig. 5). Overall, our tests show that the reconstruction algorithm allows for a robust determination of the mRNA concentration and rate of production.

We tested our reporter system by measuring the expression driven by the well-known *hb_ant* enhancer[8] (Fig. 1 and Supplementary Data 1). We used live confocal microscopy with a time resolution of 1 min per z-stack to image embryos laid from *hb_ant > mNeonRep* homozygous parents ("Methods"); the excitation laser power was optimized to obtain maximal fluorescence signal while minimizing photodegradation (Supplementary Fig. 6). As for every enhancer-reporter construct described in this study, three embryos were analysed. Starting from nuclear cycle 13, we observe a strong signal, localized mostly in the nuclei, with a pattern that recapitulates the known[13,15] spatiotemporal activity of the *hb_ant* enhancer (Fig. 1c–e and Supplementary Movie 1). To quantify the signal, we defined bins corresponding to 2% of egg length along a line connecting the anterior and posterior tips of the embryo (AP axis, 0% anterior tip, 100% posterior tip) and extended them to the cortical region, where fluorescence was measured; for simplicity, only data from the dorsal side of the embryo were used (Fig. 1e–g and "Methods"). For each bin we measured the fluorescence intensity time course and reconstructed cumulative mRNA production (Fig. 1h, i) and instantaneous mRNA production (Supplementary Fig. 7c, d). We achieved temporal registration between different datasets by visual inspection of images of the embryos that were simultaneously acquired by DIC microscopy, and defined time zero as the onset of nuclear cycle 14 (nc14). The spatiotemporal dynamics of the reconstructed RNA levels matches the known dynamics of the *hb_ant* enhancer activity (Fig. 1i): Expression begins as a broad gradient descending from the anterior tip, but expands toward the middle of the embryo later in nc14, thus creating a larger domain with a relatively sharp boundary centered around 40% AP (Fig. 1i). During the second half of nc 14, the enhancer is gradually turned off, as can be seen from the saturation of cumulative mRNA levels (Fig. 1h). The reproducibility of the mRNA reconstruction is high, with a global correlation coefficient of $r = 0.99$ for the three biological replicates (Supplementary Fig. 7b). Note that while reporter activity persists post-blastoderm, i.e., after stage 5 of embryo development, analysis of these expression patterns is difficult due to the rapid movement of nuclei that begins with gastrulation (Supplementary Fig. 8).

**The mNeonGreen reporter detects weaker expression patterns than the MS2 system**. To validate our approach, we compared our results to measurements obtained with the MS2-MCP tagging technique. We performed live imaging of an MS2-yellow reporter gene (consisting of 24 repeats of the MS2 stem loops upstream of the yellow gene coding sequence (6.4 kb)), expressed under the control of the *hb_ant* enhancer coupled to the DSCP promoter, and applied imaging conditions similar to those reported previously[13] ("Methods"). We observed the typical MS2-MCP fluorescent spots in the anterior half of the embryo (Fig. 2c, Supplementary Movie 2, and Supplementary Data 3). The strong magnification of the high numerical aperture microscope objectives that are required to detect these localized fluorescent signals limits the field of view to about 30% of the embryo length, as in previous studies[13]. However, to characterize the complete *hb_ant* activity domain it is necessary to examine larger portions of the embryos. We therefore collected data at different positions using multiple embryos. Our results agree with those reported in the literature for the *hb_ant* enhancer with the MS2-MCP system[13,15]. To compare the MS2-MCP data with the our mNeonRep results, we merged the signal from several nuclei. We find that the patterns of cumulative mRNA levels measured (MS2-MCP) or reconstructed (mNeonRep) are very similar (Fig. 2b, d), with a high correlation over all points in time and space ($r \sim 0.95$) (Fig. 2e).

We then investigated the ability of our reporter system to detect low RNA production levels. To this end, we used both techniques to measure the activity of a much weaker synthetic enhancer, consisting of three binding sites for the maternal activator Bcd, which forms a gradient extending from the anterior tip toward the middle of the embryo[25]. Previous studies used in situ hybridization to characterize the expression of this enhancer, termed *Bcd3*, and observed a weak signal in the anterior part of the embryos[25,26]. Using mNeonRep, we detect a weak but reproducible ($r = 0.83$ and Supplementary Fig. 9) gradient of reporter expression that slowly decreases from a peak at around 10% AP to up to 40% AP (Fig. 2f–g and Supplementary Movie 3). At the maximum position (arrow in Fig. 2g), the inferred cumulative mRNA level is ~30 times lower than that for the native *hb_ant* enhancer. However, it is also about 10-fold higher than background, as we verified by imaging a wild-type embryo, which is not expected to show any expression, as control (Supplementary Fig. 10). In addition, we observe a difference between the two enhancers in the dynamics of expression: the cumulative mRNA production from *Bcd3* tends to saturate earlier in nc14 than for *hb_ant*, implying that the activity of the enhancer decreases earlier (Fig. 2b, f, Supplementary Figs. 7 and 9).

Strikingly, we did not observe any clear activity domain of the *Bcd3* enhancer when using the MS2-MCP technique (Supplementary Movie 4). We could only detect a limited number of very weak fluorescents dots that are almost indistinguishable from the background noise (Fig. 2h, i, Supplementary Fig. 11b). To substantiate this observation, we compared the histogram of the intensity of detected spots for *Bcd3* with that for the *hb_ant* enhancer in posterior regions of the embryo, where this enhancer is not active (Supplementary Fig. 11c). The distributions are very similar, suggesting that the detected MS2-MCP spots for the *Bcd3* enhancer are indeed noise. We conclude that the mNeonGreen reporter system has a lower detection limit for mRNA production, making it suitable to study enhancers with low activity levels.

**Applying the mNeonGreen reporter to quantify enhancer activity**. To further illustrate the potential of our method to quantify the activity and dynamics of enhancers, we used the mNeon reporter to analyse an additional well-studied native segmentation enhancer, *Kr_CD2*, located in the *Krüppel*

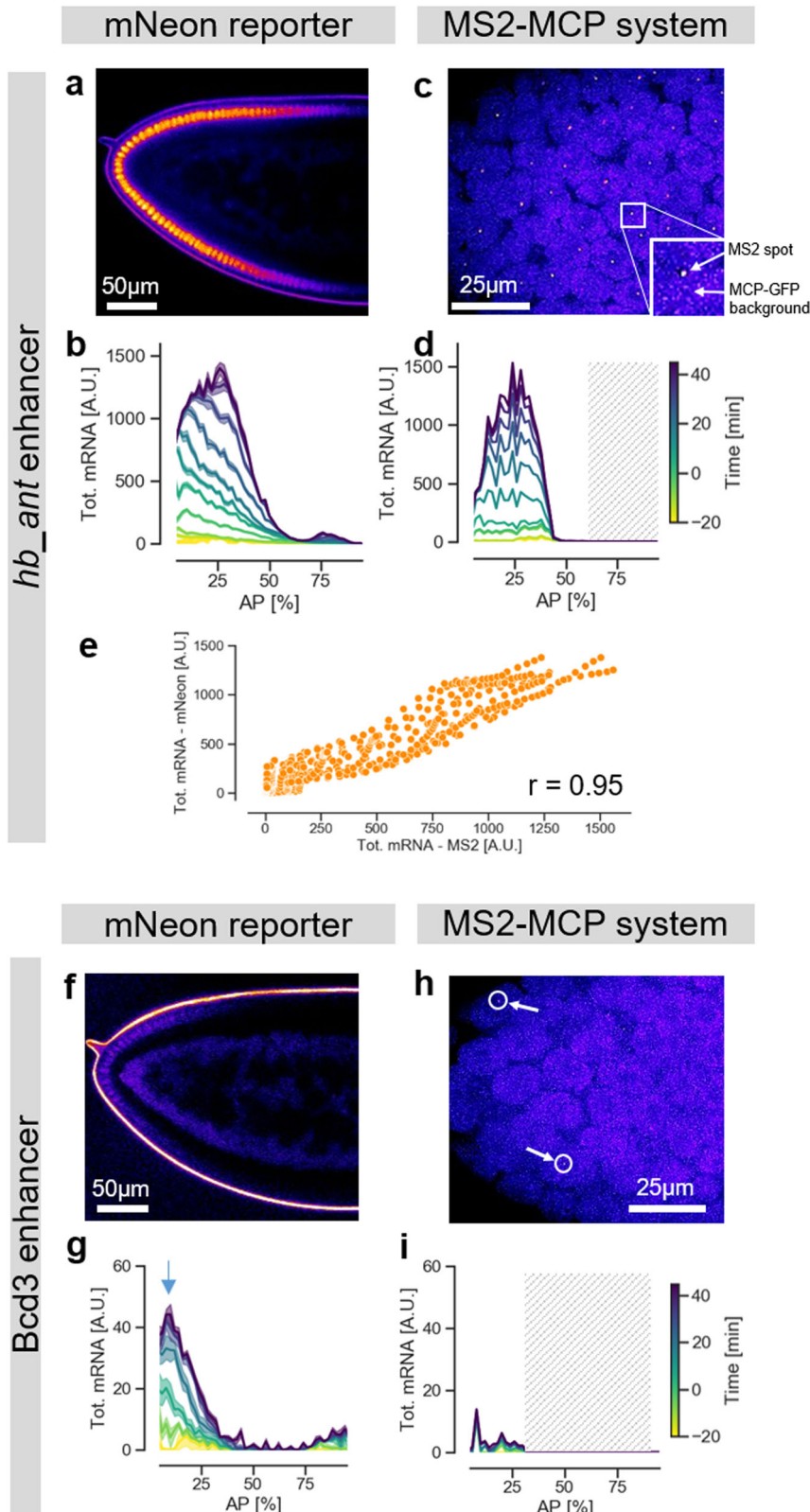

cis-regulatory region (Fig. 3, Supplementary Movie 5 and Supplementary Data 1)[8,27]. This enhancer drives expression in a narrow anterior domain extending from 0 to 25% AP and in a relatively sharp central band centered at ~50% AP (Fig. 3b, d, and Supplementary Fig. 12). The reconstructed RNA pattern matches the expression previously reported for this enhancer using in-situ stainings[8] and the MS2-MCP system[28,29], with some difference in

the size and relative strength of the anterior domain, attributable to different sequence delineation. Recent work on *Kr_CD2* has focused on the central domain, with detailed quantitative data on its dynamics generated using the MS2-MCP system[28]. Notably, our mNeonRep reporter reproduces key features of this central stripe as revealed by MS2-MCP: The stripe is well-defined and slightly narrower than the *Kr* genomic expression domain;

**Fig. 2 The mNeonGreen reporter detects weaker expression patterns than the MS2 system. a** Confocal slice of the anterior region of an *hb_ant-mNeonRep* embryo, showing mNeonGreen fluorescence in false colors **b** Cumulative mRNA production patterns at different times of embryo development measured with the mNeonGreen reporter. **c** Confocal slice of the anterior region of a *hb_ant-DSCP-MS2-yellow* embryo, MCP-GFP fluorescence represented in false colors. **d** Cumulative mRNA production patterns for the *hb_ant* enhancer as a function of AP position, measured with the MS2-MCP system. Since the field of view is limited to about 30% of the embryo, data were pooled from two independent experiments, focusing on the regions 0–30 and 30–60% AP, respectively. The shaded area represents a region of the embryo for which no data was collected. **e** Comparison of the cumulative mRNA production patterns measured with the mNeonRep and MS2-MCP reporter systems, at all times and positions. r = Pearson's correlation coefficient. **f** Confocal slice of the anterior region of a *Bcd3-mNeonRep* embryo, showing mNeonGreen fluorescence in false colors (**g**) Cumulative mRNA production patterns (maximum production level indicated by an arrow) at different times of embryo development measured with the mNeonGreen reporter. **h** Confocal slice of the anterior tip of a *Bcd3-DSCP-MS2-yellow* embryo, MCP-GFP fluorescence represented in false colors. The arrows indicate exemplary fluorescent spots. **i** Cumulative mRNA production patterns for the *Bcd3* enhancer as function of AP position, measured with the MS2 system.

expression starts only relatively late in nc14 (after $t = 15$ min, Fig. 3g)[28], and the peak of highest expression within the stripe shifts toward the anterior by about 4% of the embryo length over the time course (inset in Fig. 3d)[29]. It is also instructive to compare the temporal dynamics of the *Kr_CD2* and *hb_ant* enhancers: For both enhancers, expression in the anterior domain begins early, before the start of nc14, while expression in the central stripe of *Kr_CD2* and the weak posterior stripe of *hb_ant* initiates much later (Fig. 3e–g)[28]. This differential in the dynamics in different regions may be related to the timing of activator expression.

In a second set of experiments, we tested two variants of the synthetic *Bcd3* enhancer. We modified two independent features of the enhancer's structure that are believed to influence activity[30,31], and examined whether our experimental approach could capture any resulting differences. The first feature is the distance between the enhancer and the transcriptional start site (TSS), the second the enhancer accessibility (i.e., its chromatin state). To change enhancer-TSS distance, we removed a linker sequence between the *Bcd3* enhancer and the DSCP promoter, thereby shortening the distance between the TSS and the most proximal Bcd binding site from 146 bp to 73 bp (enhancer termed *Bcd3-proximal*). The comparison of the average expression profiles shows that the activity of the *Bcd3-proximal* enhancer is indeed consistently ~2.5-fold stronger than that of *Bcd3* (Fig. 4a, b, Supplementary Figs. 9 and 13, and Supplementary Data 3).

To test the influence of enhancer accessibility, we added binding sites for the pioneer transcription factor Zld, which is known to promote chromatin decompaction. Specifically, we embedded three Zld consensus sites in a 56 bp stretch of neutral DNA sequence[10] upstream of the *Bcd3* enhancer (Fig. 4c and Supplementary Fig. 14). Remarkably, the resulting enhancer, termed *Zld3-Bcd3*, shows a pronounced increase of reporter expression (~6 fold stronger on average) compared to the *Bcd3* enhancer (Fig. 4b, c). This finding confirms the role of Zld in increasing enhancer activity, without altering the spatial pattern[10,31].

To compare the activity profiles of the three *Bcd3* enhancer variants more directly, we plotted the ratios of cumulative mRNA production levels between *Bcd3-proximal* and *Bcd3* (Fig. 4d), and between *Zld3-Bcd3* and *Bcd3* (Fig. 4e). Interestingly, while in the first case the ratio remains constant throughout the blastoderm development, suggesting that decreasing the enhancer-TSS distance simply leads to a rescaling of activity, the *Zld3-Bcd3/Bcd3* average ratio grows over time, with values increasing from ~5 to ~7 during nc14. Thus, Zelda not only increases the rate of the enhancer activity, but also alters its dynamics. The cumulative mRNA profiles for the two enhancers indeed show that the saturation observed early in nc14 for *Bcd3* does not occur in the *Zld3-Bcd3* variant, which remains active throughout the period measured. We conclude that features that are expected to

modulate enhancer activity in similar ways, in this case by increasing the transcription rate, may result in quite different dynamics. Such subtle differences are not immediately obvious or accessible with classical reporter assays, and our experiments demonstrate the power of our mNeonRep system to capture and quantify them.

## Discussion

The mNeonGreen reporter system along with the data analysis pipeline presented in this study constitutes a valuable approach for measuring transcriptional dynamics in vivo. The optimized mNeonRep reporter has several advantages compared to the MS2-MCP system, which has been widely used for this purpose in recent years. First, our system relies on expressing only a single transgene, thus obviating the need for the complex crossing schemes required in two-component approaches and permitting speedier experimental work. Second, it provides higher detection sensitivity, as revealed by our Bcd3 enhancer, whose activity is undetectable with the MS2-MCP system, most probably due to the fluorescent background arising from unbound MCP-GFP molecules. Third, the bright mNeonGreen signal permits imaging through low numerical aperture/low magnification objectives that can capture large fields of view; this means that whole embryos can be captured at once, a distinct advantage in case of the many enhancers with broader or complex expression patterns. Taken together, these features of our reporter result in important gains in sensitivity and throughput compared to existing methods, which is particularly important if one wants to investigate enhancer function at scale and systematically test subtle modulations of enhancer composition and architecture.

It might be possible to increase the sensitivity of the MS2 system, and thus alleviate one of its limitations, through further optimization, for example by boosting the signal strength of the MS2-MCP spots by adding more repeats of the MS2 stem loops, or by using an even longer reporter gene, such that the mRNA remains sequestered at the gene locus for a longer time. This would, however, make the mRNA to be transcribed even longer than it is currently (>6 kb), which in turn would lower the time resolution with which transcriptional dynamics can be resolved. Other approaches that have successfully been used in cell culture to increase mRNA detection sensitivity, like fine-tuning MCP-GFP expression or using MCP-GFP dimers[32], might also work in *Drosophila* embryos, but they have not been tested in this paradigm.

The advantages of the mNeonRep reporter system come with modest losses in spatio-temporal resolution. While our reporter permits single-cell resolution in principle (and in other biological contexts), in the *Drosophila* blastoderm the spatial resolution is limited by the fact that the embryo at this stage is a syncytium, and in the absence of cellular membranes both the reporter mRNA and the protein can diffuse away from the site of production. However, this process is likely limited because of

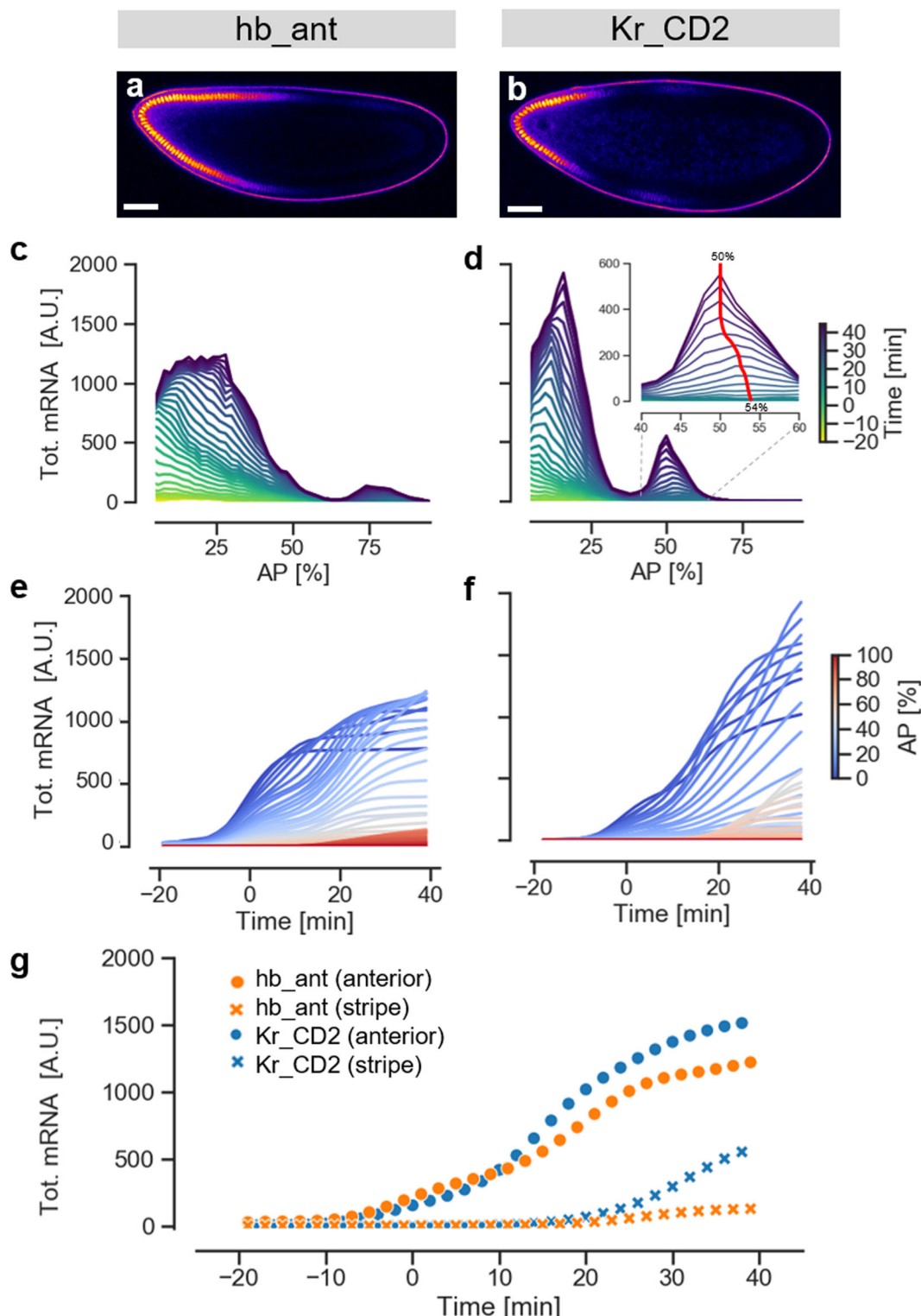

**Fig. 3 Activity and dynamics of embryos carrying native *Drosophila* enhancers using the mNeon reporter system. a** and **b**. Confocal fluorescence sections of *hb_ant* and *Kr_CD2* embryos just before gastrulation. **c** and **d**. total mRNA levels along the AP axis over time. Averages from three replicates of each enhancer. The inset in (**d**) zooms in on the evolution of the central stripe of the *Kr_CD2* enhancer. The red line highlights the dynamic shift of the position of the expression peak. **e** and **f**. Total mRNA levels as a function of time, with AP position color-coded as indicated. Averages from three replicates of each enhancer. **g** Temporal dynamics for the anterior domain (dots) and the posterior/central stripe (crosses) of the two enhancers.

the NLSs engineered into the reporter, and the resulting strong enrichment of the fluorescence signal in the nuclei. Moreover, a recent study showed that diffusion of mRNAs between neighboring nuclei in the *D. melanogaster* blastoderm is limited to only 1–2 nuclei[33]. Indeed, we find that the expression patterns obtained with our reporter, and in particular those for the narrow *Kr_CD2* central stripe, are highly similar to those measured with other techniques; this would not be the case in

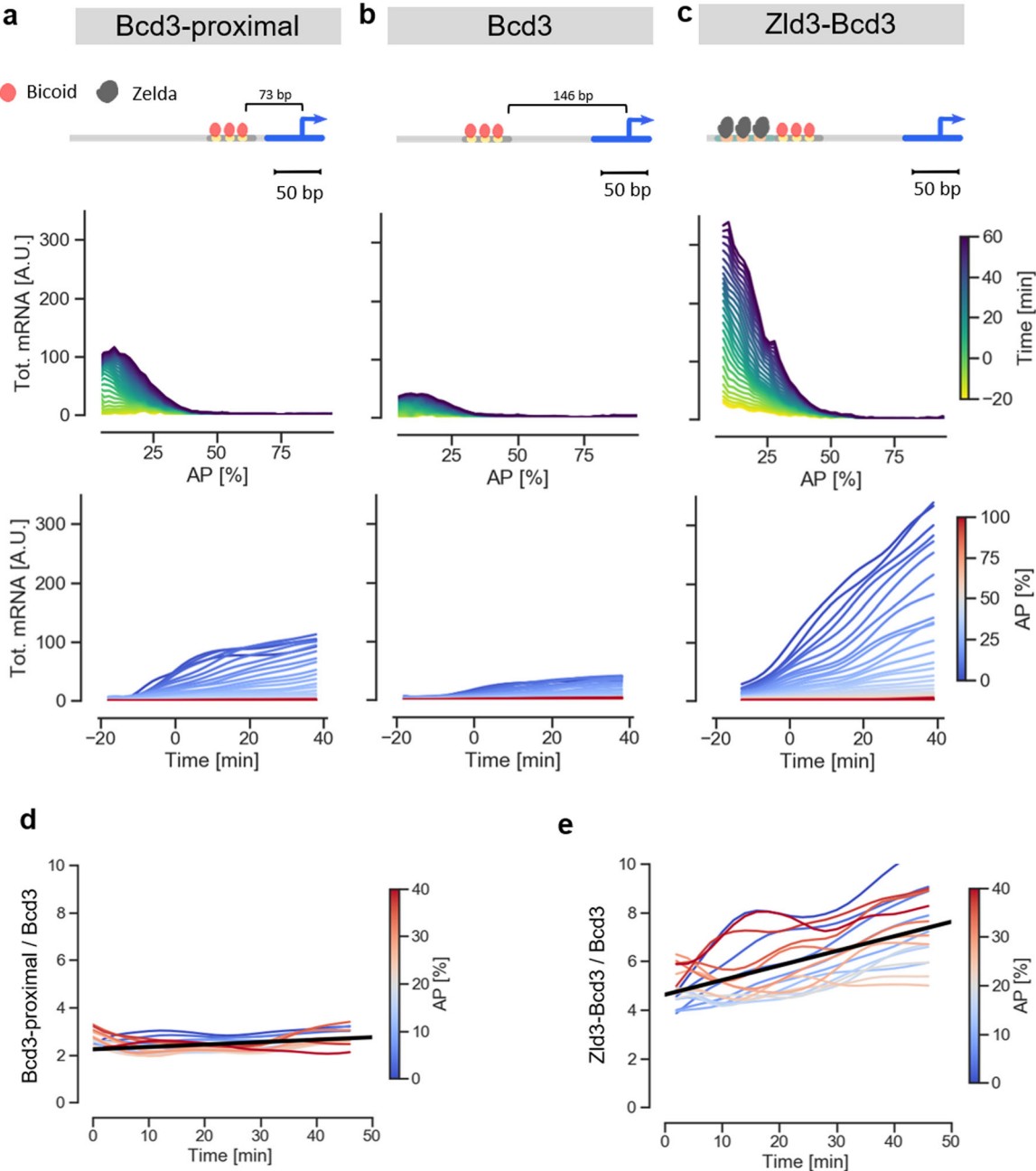

**Fig. 4 Applying the mNeonGreen reporter to measure transcriptional dynamics of synthetic enhancers. a–c** Dynamics of the activity of the *Bcd3-proximal, Bcd3* and *Zld3Bcd3* enhancers, respectively. From top to bottom: sketch of enhancer architectures, illustrating arrangement of binding sites and distance from TSS. Cumulative mRNA production patterns along the AP axis, color coded by the time of embryo development (average of 3 biological replicates). Time course of cumulative mRNA production, color coded by position along the AP axis (average of 3 biological replicates). **d** Ratio of the cumulative mRNA production for *Bcd3-proximal* and *Bcd3* enhancers as a function of time, color coded by the position along the AP axis. **e** Ratio of the cumulative mRNA production for *Zld3Bcd3* and *Bcd3* enhancers as a function of time, color coded by the position along the AP axis.

the presence of rapid diffusion of either reporter mRNA or protein.

While not measuring mRNA production rates directly, our reporter system in combination with the reconstruction algorithm is also able to capture the rapid expression dynamics characteristic of the *Drosophila* blastoderm with high fidelity. The achievable temporal resolution depends on the SNR of the protein fluorescence. Based on numerical simulations, we estimate that under our imaging conditions the reconstruction algorithm is able to resolve rapid changes in enhancer activity (e.g., a short pulse of mRNA production) with a resolution of about 7 min

(Supplementary Fig. 5). This value is modestly lower than that achievable with the MS2-MCP system (around 2–3 min), presumably because of a longer delay between RNA transcription and signal accumulation and the needed reconstruction step. However, in case of a higher SNR of the fluorescence signal, the temporal resolution of our system can be readily improved by tuning a single regularization parameter in the reconstruction algorithm (see "Methods", Supplementary Figs. 1 and 5). Given the potential cross-talk between neighboring nuclei, our reporter is not well suited to study expression dynamics in the *Drosophila* blastoderm at the single nucleus level, in particular phenomena

like stochastic transcriptional bursting, for which the MS2-MCP system is more useful. However, at slightly lower resolution the profiles of enhancer expression dynamics derived with the mNeonRep reporter are in very good agreement with those obtained through the MS2-MCP system.

To further validate and illustrate the performance of our mNeonRep reporter, we used it to investigate the differences between three variants of a simple synthetic enhancer, consisting of binding sites for the maternal activator Bcd. Although near identical in their spatial expression pattern, the three enhancers show clear differences both in their absolute levels of activity and in their temporal dynamics, thus illustrating the ability of the reporter to capture subtle quantitative differences as well as their development over time. They also provide interesting biological insight. First, the *Bcd3-proximal* enhancer shows that activity increases when the enhancer is placed closer to the TSS. While enhancer activity was originally thought to be independent of the orientation and distance to the core promoter[34], mild effects of locus architecture, and in particular of enhancer-promoter distance, on enhancer function have been reported[35,36]. In the simpler case of bacterial enhancers, a systematic analysis of synthetic enhancers positioned at a range of distances from the promoter (20–500 bp) revealed a non-linear dependency, with greatest activity at a distance of around 70 bp[30]. Our findings confirm such a dependency, which may reflect the probability of contacts between enhancer and core promoter, for *Drosophila*. Note that changing the enhancer-promoter distance tunes the level of expression without altering dynamics: both *Bcd3* and *Bcd3-proximal* show a rapid reduction of transcription early in nc14, and throughout the time course and at all positions, the *Bcd3-proximal* enhancer behaves simply like a rescaled version of the *Bcd3* enhancer.

By contrast, adding three binding sites for the pioneer transcription factor Zld[37] to the *Bcd3* enhancer not only dramatically boosts activity levels, but also changes the dynamics of expression: Unlike *Bcd3* and *Bcd3-proximal*, the activity of *Zld3-Bcd3* remains sustained throughout nc14, with steadily increasing cumulative levels of mRNA, and an increasing activity ratio relative to *Bcd3* at all positions. Binding of Zld to its cognate sites promotes the deposition of histone modifications and increases DNA accessibility[38], and inserting binding sites for Zld in native or synthetic enhancers has been reported to increase enhancer activity and accelerate enhancer activation after nuclear divisions[10,31]. More specifically, single molecule studies have shown that Zld creates local hubs of increased Bcd concentration in the vicinity of enhancers, thus increasing site occupancy[39]. The increase in absolute activity resulting from the insertion of Zld sites into our *Bcd3* enhancer is therefore expected. The change in expression dynamics is more intriguing: Since the decrease in Bcd-dependent transcription early in nc14 is thought to result from the sumoylation of the Bcd protein[40,41], the sustained activity in the presence of Zld sites could result from a threshold effect, with greater enhancer accessibility boosting the ability of low levels of remaining Bcd to activate transcription, or it could be the result of a more direct interference of Zld with Bcd deactivation, for example by sequestering Bcd in a local microenvironment at the enhancer where sumoylating enzymes are not present.

In summary, we have shown that our mNeonGreen reporter system is a powerful tool to study transcriptional dynamics, and is particularly suited for studies that aim to quantify expression dynamics of large numbers of native or synthetic enhancers, including constructs with weak expression levels. Compared to existing methods, our approach offers important advantages in terms of sensitivity and throughput, and it can, with suitable adaptations, be applied in other organisms or cell cultures.

## Methods

**Cloning of transgenes**. The mNeonGreen reporter construct was generated by C- and N-terminal fusion of a codon optimized (*Eurofins genomics GENEius* software package - Munich, Germany) mNeonGreen[17] coding sequence, obtained by gene synthesis, to three different NLSs: the Bipartite-N-term NLS[42], the SV40 NLS, and a Class3 C-term NLS[21]. All enhancers were coupled to a strong synthetic core promoter (DSCP)[43]. The sequence of *hb_ant* enhancer was amplified from genomic DNA. The sequence of the *Bcd3* enhancer[25] was generated by oligo annealing. The sequence containing 3 Zld binding sites was designed by inserting three consensus binding sites for Zld[44] into a neutral background described in previous works[10], and was generated by oligo annealing. In all constructs, except for Bcd3-proximal, a 73-bp linker separated the enhancer from the basal promoter. This sequence does not contain any predicted binding site for TF of the segmentation network, based on available position weight matrices (PWM)[44]. To optimize translation of the construct, we included the IVS + Syn21 translational enhancer sequences[22] at the 5'UTR and the p10 terminator sequence[22] at the 3'UTR. We refer to the full sequence of the reporter including 3'UTR and 5'UTR as mNeonRep. All elements were cloned into an expression construct based on the pBDP backbone (a gift from Gerald Rubin; Addgene plasmid #17566) as described in[45] with only one difference: the insertion of an additional 340 bp long neutral spacer[46] upstream of the enhancer. This was done in order to minimize the potential interference by binding sites for segmentation TFs in the backbone of the reporter plasmid, which were revealed by a preliminary analysis.

For the generation of the MS2 reporter construct, the 24XMS2 tag (a gift from Robert Singer, Addgene plasmid #31865) was fused upstream of the yellow reporter gene coding sequence (a gift from Liqun Luo, Addgene plasmid #24350). The 24xMS2-yellow sequence was then cloned immediately downstream of the enhancer-linker-DSCP sequence into the same pBDP backbone described above for the mNeonGreen reporter construct. A complete list of all sequences is provided in Supplementary Information.

**Fly stock generation**. All reporter plasmids for both the mNeonGreen and MS2 reporters were integrated in the same attP2 docking site using PhiC31 integrase[43]. Homozygous fly stocks were generated by crossing a single male with a single homozygous virgin female, and the insertion of the correct construct was verified by single-fly PCR of both parents and sequencing of the PCR products.

**Live Imaging**. *mNeonGreen*: Enhancer-mNeonRep embryos were collected, dechorionated in 50% bleach and mounted between a semipermeable membrane and a microscope cover glass, immersed in halocarbon oil (Sigma). Imaging was performed at 24 ± 1 °C on a Zeiss LSM710 confocal microscope using a 40×1.2NA water immersion objective. The laser excitation power was chosen to provide maximal fluorescence intensity while keeping photodegradation negligible: with our setup, we found that the optimal excitation, measured at the entrance pupil of the objective was 8 μW (Supplementary Fig. 6). Pixel size was set to 1.1 μm. Two tiled stacks, each consisting of 3 images separated by 7.5 μm in z, were acquired at each time point. The resulting field of view of 250 μm × 580 μm allowed us to image an entire embryo in a single movie with a time resolution of 60 s per z-stack.

*MS2-MCP*: yw;Histone-RFP;MCP-GFP (Bloomington *Drosophila* Stock center #60340) virgins were crossed with males carrying either the *hb_ant*-DSCP-MS2-Yellow or *Bcd3*-DSCP-MS2-Yellow reporter genes. Embryos were collected and mounted as described above. Imaging was performed on a Zeiss LSM710 confocal microscope using a 63 × 1.4NA oil immersion objective. Pixel size was set to 0.33 μm and the image field of view to 169 μm x169 μm. A stack of 15 images separated by 1.3 μm in z was acquired at each time point. The final time resolution was 60 s per z-stack. At the end of each movie, a single tiled z-stack with a much larger field of view of 500 μm × 840 μm was acquired to precisely locate the imaged area within the whole embryo.

**Alfa-Amanitin and cycloheximide injection**. To measure the degradation rate of reporter RNA in the embryo, transcription was blocked by injecting alpha-amanitin (Sigma-Aldrich: A2263) using a needle, at the concentration of 0.4 mg/ml[47]. To measure the maturation rate of fluorescence of the reporter protein in the embryo, translation was blocked by injecting cycloheximide (Sigma-Aldrich: 01810) using a needle, at the concentration of 0.9 mg/ml[48]. In both cases, Enhancer-mNeonRep embryos were collected, dechorionated in 50% bleach, glued on a coverslip and dried at 18 °C for 20 min prior to the injections. Injected embryos were imaged as described above; for each condition, three replicates were measured, with nearly identical results.

**Image segmentation**. Confocal stacks of embryos were processed as follows, using the *Definiens XD 2.0* software package (Munich, Germany). In brief, the contour of the embryo was first identified using the strong autofluorescence signal arising from their vitelline membrane. The cortical region containing the nuclei was then defined by shrinking the contour line of the embryo twice by a few pixels toward its interior (region delimited by the green lines in Fig. 1e). We then applied a watershed segmentation to produce small segmented patterns, randomly distributed and of various sizes in the segmented cortical region (typically 6–10 pixels; each individual segmented object corresponded typically to less than 1% of the

embryo length). For each z-slice, mean mNeonGreen fluorescence signal was then computed for each object. Finally, individual segmented objects were pooled into bins corresponding to 2% egg length, and their size-weighted average mNeonGreen fluorescence intensity was computed.

### Signal linearity, background subtraction and correction for photobleaching.
An important assumption in quantitative fluorescence live imaging is that the recorded fluorescence signal is directly proportional to the fluorophore concentration. We checked the validity of this assumption under our imaging conditions using a serial dilution of Rhodamine6G in a 96 well plate. We found that the response of our setup is non-linear only at very low concentration (Supplementary Fig. 15). The observed non–linearity can be described by a polynomial fit, which we used to correct all raw data.

The fluorescence signal measured in the embryo cortex arises from two independent contributions: the reporter protein fluorescence that we aim to isolate, and the embryo autofluorescence. Moreover, while under our imaging conditions the mNeonGreen signal is not affected by photobleaching, the background autofluorescence is, and thus changes over the measurement time course. In order to estimate the background autofluorescence, we computed the average background profile along the antero-posterior axis by averaging the signal of 4 frames at the beginning of each movie, when no mNeonGreen fluorescence is yet detectable. To account for photobleaching, we measured the time course of the signal in a region of the embryo where no expression of the reporter protein was observed, and then used this estimate to rescale the average background at each time point, assuming the embryo autofluorescence to be equally affected by photobleaching at all positions in the embryo.

### Temporal registration.
In order to compare expression patterns between different embryos it is important to precisely register them in time relative to a common reference point. We chose this time point (time zero, $t = 0$) to be the moment at which nuclear membranes reappear throughout the embryo after mitotic division following NC13, and used DIC images to detect this. Time zero was determined with a precision of ±1 min, which also matched the time resolution of our measurements.

### Analysis of reporter time course and mRNA reconstruction.
Protein levels offer an indirect readout of transcription. To derive the underlying mRNA production rates, we followed the approach presented in[24] and we modeled the expression of our reporter with a set of ODEs:

$$\frac{dM(t)}{dt} = mRNA_p(t) - k_{dm}M(t), \tag{1}$$

$$\frac{dD(t)}{dt} = k_pM(t) - (k_{dp} + k_{mat})D(t), \tag{2}$$

$$\frac{dF(t)}{dt} = k_{mat}D(t) - k_{dp}F(t). \tag{3}$$

The model includes the concentration of reporter mRNA $M(t)$, the concentration of nonmature dark reporter protein $D(t)$, and the concentration of the mature fluorescent protein $F(t)$. Maturation of the protein and degradation of both protein and mRNA is accounted for by the rates protein maturation $k_{mat}$, protein degradation $k_{dp}$, and mRNA degradation $k_{dm}$. Using the linearity of this model and discretizing the time, this system of differential equations can be rewritten as a linear model. The linear model links, through a model matrix H, the array of observed fluorescence $F(ti)$ to the mRNA production rate $mp(tj)$ and to the initial concentrations of mRNA m(0), immature and mature protein P(t0) and F (t0).

$$F = \begin{bmatrix} F(t_0) \\ \vdots \\ F(t_n) \end{bmatrix}, m_p = \begin{bmatrix} mRNA_p(t_0) \\ \vdots \\ mRNA_p(t_m) \\ M(t_0) \\ P(t_0) \\ F(t_0) \end{bmatrix} F = H\,m_p. \tag{4}$$

The matrix H depends only on the structure of the original ODE system and on the rates of mRNA degradation $k_{dm}$, protein degradation $k_{dp}$, and protein maturation $k_{mat}$.

The linear model can be fit to the time course of protein fluorescence $F(ti)$ using a regularized non-negative least square algorithm which determines the rate of RNA production minimizing:

$$\|F - H\,m_p\| + \lambda \sum_0^{m-1} \|m_p(t_m) - m_p(t_{m+1})\|, m_p > 0, \tag{5}$$

under the assumptions that the mRNA production rate and all concentrations must be positive definite. After fitting we calculated the cumulative mRNA

production as:

$$mRNA_{Tot}(t_i) = \sum_{j=0}^{i} m_p(t_j). \tag{6}$$

In order to estimate how sensitive to the noise the reconstruction is, we implemented a bootstrapping of residuals algorithm. We first checked that the residuals of the fit showed no correlation at different times (Supplementary Fig. 16a), and that the standard deviation of the residuals scaled linearly with the intensity of the signal (Supplementary Fig. 16b). Then, we implemented bootstrapping by rescaling the residuals with the average signal, reshuffling the normalized residuals and rescaling them back again (Supplementary Fig. 1). Using this protocol, we built a "synthetic" time course of fluorescence and, by fitting the data, we obtained a set of predicted mRNA production rates. Since the procedure can be repeated N times, we can use it to obtain a population of predicted mRNA production rates at each time, from which we derive confidence intervals.

The RNA reconstruction relies on a regularized linear regression algorithm which includes a smoothing parameter $\lambda$. Decreasing the value of this parameter increases time resolution (Supplementary Fig. 5d). However, too small values of $\lambda$ give rise to unstable reconstructions creating artifacts and large errors in the estimates (Supplementary Fig. 5e). In this respect, bootstrapping also offers an internal control that guides the choice of $\lambda$. The minimum value of $\lambda$ that gives reproducible reconstructions depends on the signal-to-noise level of the measured protein fluorescence. A higher SNR allows for obtaining reproducible results with lower values of $\lambda$ and thus increases temporal resolution.

### Analysis of mNeonGreen reporter maturation time and mRNA degradation rate.
To measure the maturation rate of mNeonGreen and the degradation rate of its mRNA, we analyzed the data of cycloheximide or alpha-amanitin injected embryos. To interpret these datasets, we used the model described by Eq. (4). Injection of cycloheximide blocks protein translation and can be described by setting $k_p = 0$. Under this assumption the concentration of fluorescent protein is expected to follow:

$$F(t) = Ae^{-k_{dp}(t-t_0)} - B(e^{-k_{mat}(t-t_0)} - 1)e^{-k_{dp}(t-t_0)}. \tag{7}$$

Since at the timescales of our measurements mNeonGreen degradation turns out to be negligible (fluorescence reaches a stable plateau in Supplementary Fig. 2d) we can also set $k_{dp} = 0$ and further simplify this relation to:

$$F(t) = A - B(e^{-k_{mat}(t-t_0)} - 1). \tag{8}$$

Fitting this equation to the fluorescence time course of cycloheximide-injected embryos, we inferred the values of mNeonGreen maturation rate in the embryo (Supplementary Fig. 2d, e).

To analyze the data obtained from alpha-amanitin injected embryos, we set $mRNA_P = 0$ in *Equation* and obtained:

$$F(t) = \frac{A\left(e^{k_{mat}t}\left((k_{dm} - k_{mat})e^{k_{dm}t} + k_{mat}\right) - k_{dm}e^{k_{dm}t}\right)}{k_{dm}(k_{dm} - k_{mat})e^{t*(k_{dm}+k_{mat})}} + B\left(1 - e^{-k_{mat}t}\right) + C. \tag{9}$$

By fitting this equation to the time course of fluorescence, we inferred the value of the reporter mRNA degradation rate in the embryo (Supplementary Fig. 2f, g).

### Analysis of MS2 data.
MS2 fluorescent spots were detected in 3D with *Definiens XD 2.0*. We first identified the cortical region of the embryo, in the manner described above. We then segmented in 3D the transcription foci exhibiting accumulated GFP fluorescence light with the following procedure. The considerable variation in background intensity makes a precise identification of GFP-labeled foci with weak signal intensity difficult. Therefore, application of a global threshold does not produce good segmentation. Instead, we developed a strategy based on background reduction. As a first step, we applied a 3D-Gaussian filter with a kernel size of $3 \times 3 \times 3$ pixels (GaussianFilter1), then applied a second 3D-Gaussian filter, again with a kernel size of $5 \times 5 \times 3$ pixels, (GaussianFilter2). We then subtracted GaussianFilter2 from Gaussianfilter1, which resulted in a background subtracted image. As a last step, we applied a global threshold and carried out segmentation using a so-called Multi-Threshold Segmentation algorithm implemented in the *Definiens* software platform. Briefly, this algorithm splits the image domain and classifies the resulting image objects based on a defined pixel value threshold. To reject particles resulting from segmentation errors, we filtered the resulting image objects by setting a threshold based on particle volume. Hence, with this procedure we avoid to include in our data artefacts due to detector shot noise, fluctuations of GFP background, or embryo autofluorescence.

Following a previous study[13] we calculated the MS2 signal as the integral of the fluorescence of each particle minus the local GFP background that was estimated from a spherical shell surrounding each detected spot. To compensate for differences in imaging depth, the signal was further normalized by the average local background.

To calculate the expression profiles, the data were binned based on their AP position, by using 2% AP bins. We defined the total mRNA production rate in each bin as the sum of MS2 signal in the bin. This allowed for integrating both the information on the fraction of active nuclei and on the intensity of transcription in each cell. Since the imaged portion of embryo at different AP position is not

uniform (Supplementary Fig. 11a, b), we normalized the total mRNA production rate in each bin by the width of the embryo at the respective AP position. At last, we defined the total mRNA production as the integral over time of the mRNA production rate inside each bin.

**Statistics and reproducibility**. To assess the reproducibility of the measurements performed with the mNeon reporter, we measured three biological replicates. Each replicate is an independent measurement of a different embryo of the same stock. The confidence intervals on the reconstructed mRNA level of each replicate have been calculated by bootstrapping, details about the bootstrapping procedure are provided in "Methods" section: "Analysis of Reporter time course and mRNA reconstruction." Where data from a single replicate are shown with a confidence interval, the interval represents the first and third quartile.

**Reporting summary**. Further information on research design is available in the Nature Research Reporting Summary linked to this article.

## Data availability
No publicity datasets were used. All figures have associated raw data and there is no restriction about data availability. All raw time courses of fluorescence intensities and extracted mRNA levels are available as Supplementary Data 1–3 for the natural enhancers, the MS2 measurements, and the synthetic enhancers, respectively.

## Code availability
All Definiens and *Python3.7* codes used for image and data analysis are available author upon request to the corresponding author.

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

## Acknowledgements

We dedicate this publication to the memory of Prof. Ulrike Gaul who passed away after a long illness during the review of the manuscript. This work was supported by the Deutsche Forschungsgemeinschaft through Sonderforschungsbereich SFB 646 and the Center for Integrated Protein Science Munich (CIPSM). S.C. and M.B. were supported by a DFG Fellowship through the Graduate School of Quantitative Biosciences Munich (QBM). U.G. acknowledges support by the Humboldt-Foundation (Alexander von Humboldt-Professorship). We are grateful to Prof. David Arnosti and Prof. Don Lamb for stimulating discussions and comments on our research work. We thank F. Mura for proofreading the manuscript and M. Schnepf and all members of the Gaul lab for valuable discussion.

## Author contributions

U.G. developed the project; S.C., C.J., and U.G. designed the experiments; S.C., M.B., and A.E.S. designed and produced the synthetic enhancer sequences; S.C. and M.H. performed the experiments; S.C., U.U., and C.J. performed data analysis; S.C., N.G., U.U., and C.J. wrote the manuscript.

## Funding

## Competing interests

The authors declare no competing interests.
