## [Peer Review File · Communications Biology]

Reviewers' comments:

Reviewer #1 (Remarks to the Author):

In this paper, Stefano Ceolin and co-workers have engineered an expression reporter to monitor the activity of specific promoters during the development of *D. melanogaster* embryos. Their reporter is based on the mNeonGreen fluorescent protein coupled to multiple NLS, a strong promoter and a translation activator. Specific enhancers can be placed upstream of this construct in order to monitor the spatial and temporal regulation of these enhancers during embryonic development. A mathematical model was developed to translate the fluorescent signal measured into an expected mRNA production level. In addition, their sensor is compared to the MS2 assay that allows a direct detection of the mRNA production. Overall the authors show that the mNeonGreen reporter provides a better sensitivity than the MS2 assay without sacrificing much in terms of temporal resolution in gene expression dynamics.

Overall, the paper is nicely written and well illustrated. While the approach is not completely novel, the careful design and calibration of the reporter in the settings of the fly embryo makes it a useful tool for the community. I have one concern with this study and a number of smaller technical comments.

While I agree with the general message of the paper which is that rapidly maturing fluorescent reporters can offer an alternative to transcriptional reporters by offering a better sensitivity, my fear is that the experimental set-up used to measure the MS2 signal may not be optimal and doesn't do justice to this assay. One important factor controlling the sensitivity of this method is the complete length of the induced transcript. Indeed, a longer transcript leads to a larger number of polymerase accumulating on the transcribed locus and thus this results in a stronger signal from the transcription site. In their original study, Garcia et al (Cur Biol, 2013) placed the LacZ gene downstream of the 24 MS2 stem loop providing an additional ~4kB of transcribed mRNA. In the current study, the authors don't specify the length of the transcribed locus placed downstream of the stem loops. Another factor controlling the sensitivity of the MS2 assay is the level of expression of the phage coat protein. Would a decrease in expression of the MS2-GFP construct allow a detection of the transcription sites in conditions where the transcript is less expressed? The authors should discuss these possible optimizations of the MS2 assay in the text.

Technical Comments

The authors claim that the mNeonGreen signal is not affected by photo-bleaching under their conditions. How did they evaluate this?

They use a complex method to remove the contribution of the autofluorescence background to the mNeonGreen signal. Did they perform a control with a non-fluorescent embryo to test the efficiency of their approach?

In the cycloheximide and alpha-amanitin control experiments, it is not clear to me how the time zero is defined. In panel d of Sup Figure 2, the red traces start 10min after time zero. How is this delay identified? Where does this delay come from? Is there a delay caused by the slow action of cycloheximide in the embryo? Were these control experiments performed in triplicate?

In Sup Figures 3 and 4, the modeled mRNA production always deviates from the true production rate at late time points. Why is this happening? How does this error affect the analysis of the reporter measurements?

Line 427: the labeling of the rate constant k_m and k_{dm} is inverted.

Line 524: Equation number is missing.

Reviewer #2 (Remarks to the Author):

Ceolin and colleagues submitted the manuscript titled "A sensitive mNeonGreen reporter system to measure transcriptional dynamics in *Drosophila* development" to Communications Biology to be peer reviewed for publication. This journal is "an open access journal that publishes high-quality research from all areas of the natural sciences. Papers published by the journal represent important advances of significance to specialists within each field."

In animals, genes experience regulation at the level of transcription control. Regulation comes from spatial, temporal, and quantitative inputs. Much of this regulation occurs at enhancer elements. Fluorescent Reporter Genes when coupled to an enhancer has been a powerful method to study the activity of enhancers and identify the functional sequences within enhancers. However, the standard fluorescent reporter has some limitations, notably at the level of temporal regulation. The authors here optimized the bright and fast-maturing fluorescent protein mNeonGreen as a real-time, quantitative reporter of enhancer expression. Using a synthetic enhancer (and variant forms of this enhancer) active during *Drosophila* embryonic segmentation the authors derived enhancer activities from the reporter fluorescence dynamics with high spatial and temporal resolution, using a robust reconstruction algorithm and established correlations with the expected levels of RNAs.

The methods and results seem rigorous and their description was clear and thorough. However, the results provided failed to inspire a belief that this system was sufficiently vetted to support it being adopted by others in the transcription regulation community. I suggest expanding this study and making a new more expansive manuscript, or selecting a more appropriate journal for publication.

Major Concerns:

This system, although shown to be excellent in a specific context, is not likely to draw a modestly broad readership. The manuscript focused on synthetic enhancer elements in very specific embryo context. Demonstration of a more broad utility for this would improve the manuscript. For example, how does it work with multiple non-synthetic enhancers and for enhancers active at additional developmental ages.

Minors Concerns:

Figure 1 panels c-e please annotate the identity of the enhancer driving reporter expression.

Figure 1, if EL% and AP% are the same thing, please pick one and stick to it.

Figure 2 has red squiggly line in label. Please remove.

Figure 2 legend. What is the arrow (it is stated in text, but not in the figure legend)? Change "data are pooled" to "data was pooled"

Line 97 and 107. The days of "data not shown" are over. Please add to supplement or else do not mention.

Line 140. Remove "/"

Line 195. Remove "a"

Figure 3a, scale bar is missing size in base pairs.

Reviewer #3 (Remarks to the Author):

In this manuscript, authors have used fast-maturing fluorescent protein mNeonGreen as a quantitative reporter of enhancer expression. With several experiments, they have proved that this reporter has higher sensitivity than the MS2-MCP system and can be effectively used to measure the strength of enhancers.

My comments are appended below:

1. What is the half-life of the reporter? For being a real-time reporter, fluorescent protein should be fast-degrading along with fast-maturing. It would be very informative if authors can show how long does it take the signal to turn off after the enhancer is silenced?
2. Further, a PEST domain can be added to the reporter to reduce its half-life. Although, this is required only if the half-life of the reporter is very long and signals are seen even if the enhancer activity is turned-off.
3. The authors have used the hb_ant enhancer to show that the mNeonGreen signal is localized along with the spatiotemporal domain of hb_ant. The expression domain of hb_ant is very wide, so

to confirm spatial restriction of the reporter it will be better to use enhancers having a smaller expression domain like pair-rule enhancer or any segment-polarity enhancer that expresses in segments.

4. Authors have provided conclusive evidence that the reporter can be used to check the activity of weak enhancers like Bicoid which proves that the reporter is very sensitive.

5. Also, authors have shown that adding Zfd binding sites can increase the activity of weak enhancers and hence has the potential to facilitate the study of weak enhancers.

Reviewers' comments:

Reviewer #1 (Remarks to the Author):

While I agree with the general message of the paper which is that rapidly maturing fluorescent reporters can offer an alternative to transcriptional reporters by offering a better sensitivity, my fear is that the experimental set-up used to measure the MS2 signal may not be optimal and doesn't do justice to this assay. One important factor controlling the sensitivity of this method is the complete length of the induced transcript. Indeed, a longer transcript leads to a larger number of polymerase accumulating on the transcribed locus and thus this results in a stronger signal from the transcription site. In their original study, Garcia et al (Cur Biol, 2013) placed the LacZ gene downstream of the 24 MS2 stem loop providing an additional ~4kB of transcribed mRNA. In the current study, the authors don't specify the length of the transcribed locus placed downstream of the stem loops.

We thank the Reviewer for the opportunity to clarify this very important point. We agree that for a fair and rigorous comparison with the MS2 system, the length of the transcribed gene and the number of the MS2 stem loops should be the same as what previous studies have used. This was indeed the case in our experiments since, following the most recent MS2 studies in *Drosophila* embryos, we used 24 MS2 stem loops upstream of the long *yellow* reporter gene (6.4kb) for the generation of the MS2 reporter constructs used in our study. This was not clearly stated in the main text of the manuscript but only indicated in the **METHODS** section (page 20 at line 367). To make it clearer, we replaced in the main text on page 10 at line 172 (manuscript before revisions):

“..... We performed live imaging of an MS2-yellow reporter gene expressed under the control of the *hb_ant* enhancer”

with

“..... We performed live imaging of an MS2-yellow reporter gene, consisting of 24 repeats of the MS2 stem loops upstream the yellow gene coding sequence (6.4kb) expressed under the control of the *hb_ant* enhancer”

Another factor controlling the sensitivity of the MS2 assay is the level of expression of the phage coat protein. Would a decrease in expression of the MS2-GFP construct allow a detection of the transcription sites in conditions where the transcript is less expressed? The authors should discuss these possible optimizations of the MS2 assay in the text.

A decrease in expression of the phage coat protein MCP-GFP would still allow a detection of the transcript as long as the MCP-GFP is co-expressed in large excess compared to the MS2-RNA transcript. This is required to achieve a rapid binding of MCP-GFP to the nascent transcripts. The expression level of the MS2-*yellow* gene transcripts can indeed strongly weaken detection for a weak enhancer, as it is the case for the Bcd3 enhancer used in this work. The limit in sensitivity is mainly due to the strong fluorescent background arising from the unbound MCP-GFP molecules and to its local fluctuations which give rise to spots of fluorescence that could mistakenly be detected as nascent transcripts thus limiting dramatically the signal-to-noise ratio. It is therefore not clear that reducing MCP-GFP expression would provide a substantial increase in sensitivity and fine-tuning the expression level in *Drosophila* embryos would be quite laborious. However, it's also true that alternative strategies improved mRNA detection efficiency of the MS2-MCP system in cell cultures, which could be applied in the embryo as well. Following the Reviewer advice, we discuss possible optimizations of the MS2 assay in the main text by adding in the discussion on page 16 at line 263:

“Note that a possible way of enhancing the fluorescence signal emitted by the MS2-MCP spots would be the addition of more repeats of the MS2 stem loops, or the use of an even longer reporter gene, such that the mRNA remains sequestered for a longer time at the gene locus. This would, however, make the mRNA to be transcribed even longer than it is already (>7000 bps), which would lower the time resolution with which transcriptional dynamics can be resolved. In principle, other approaches that have successfully increased mRNA detection sensitivity in cell culture, like fine-tuning MCP-GFP expression or using MCP-GFP dimers (Wu et al., *Biophys. J.*, 2012), could also improve the sensitivity of the MS2-MCP systems in *Drosophila* embryos, but have not yet been tested in this system.”

Technical Comments

The authors claim that the mNeonGreen signal is not affected by photo-bleaching under their conditions. How did they evaluate this?

In pre-experiments that were originally not shown in the manuscript, we evaluated the effect of photobleaching by acquiring time series of *hb_ant*-mNeon green embryos. In all live-imaging experiments presented in the manuscript, we chose a laser excitation power that provides maximal fluorescence intensity while keeping photodegradation negligible: with our setup, we found that the optimal excitation, measured at the entrance pupil of the objective was 8 μ W. To help researchers to adapt and optimize the mNeon reporter for their systems, we added a new supplementary figure that illustrates how to best choose the excitation laser power to maximize signal and minimize photobleaching (**Supplementary Fig. 5**). We mention this in the main text by adding on page 9 at line 148:

“The excitation laser power was optimized to obtain maximal fluorescence signal while ensuring negligible photodegradation (**Supplementary Fig. 5**)”

We also added in **METHODS** which laser power we selected and added a reference to **Supplementary Fig. 5**.

Note that since transcription and translation of our mNeon reporter occurs continuously in a living embryo, one cannot directly infer that photobleaching is negligible from a movie of a developing embryo. However, in cycloheximide injected embryos, translation is stopped and no

further mNeon reporter molecules are produced. **Supplementary Fig. 2d** (and the plots of new replicate measurements shown below) display the time course of the fluorescent intensity in embryos where one can see that once translation is stopped (at about $t=30\text{min}$) the fluorescence intensity remains constant until the end of the measurements (plateaus of the red and orange curves). This demonstrates that photobleaching is indeed negligible under our imaging conditions. However, these measurements lasted for only ~ 50 min, since gastrulation occurs afterwards.

For another confirmation of the negligible effect of photobleaching, we imaged two non-injected embryos carrying the *hb_ant* enhancer for over 5 hrs, with one frame per minute (our usual condition) and one frame every 20 mins, respectively. The fluorescence time courses of the two datasets (**Supplementary Fig. 5b**) overlap very well, demonstrating that photobleaching is very low. The shapes of the curves are explained in more details below when presenting the new **Supplementary Fig. 7** in the answer to Reviewer #3.

They use a complex method to remove the contribution of the autofluorescence background to the mNeonGreen signal. Did they perform a control with a non-fluorescent embryo to test the efficiency of their approach?

The heterogeneous autofluorescence background can be estimated from very early embryonic stage before the first transcription bursts or from the posterior part of Hb-enhancers embryos during transcription since there is no expression of mNeon in the posterior. This procedure to determine the heterogeneous background presents the advantage to account for biological variation of the background between different embryos. However, as an additional control we measured a wild-type embryo as suggested by the Reviewer (**Supplementary Fig. 9**). Whereas a weak autofluorescence can be observed in the yolk, the background signal is extremely weak in the in the cortical region of the embryo where the nuclei are. We analysed confocal images from the wild-type embryo using the same analysis pipeline applied to embryos expressing the mNeon reporter, including image segmentation, background correction and mRNA reconstruction. The final result doesn't show any evident artefact or systematic error introduced by the data analysis pipeline. Moreover, this analysis allows to characterize the noise level of our method which corresponds to ~ 5 A.U. of total mRNA, this level is 10 times lower than the expression level of the weakest enhancer measured in this study. We added on page 11 at line 194 of the main text:

“As an additional control, we imaged a wild-type embryo (*wt*), which as expected didn't show any expression (**Supplementary Fig. 9**). The total mRNA level reconstructed from the fluorescence signal in the anterior part of *Bcd3* was about 10-fold higher than that obtained for the *wt*.”

In the cycloheximide and alpha-amanitin control experiments, it is not clear to me how the time zero is defined. In panel d of Sup Figure 2, the red traces start 10min after time zero. How is this delay identified? Where does this delay come from?

For all confocal time series of this work the time zero is clearly defined (within ca 30s) by the beginning of nc14, which can be clearly seen in DIC images of the embryos (that we always acquired together with the fluorescence images). However, in **Supplementary Fig. 2d** the red traces start only after 10 min because of the additional time delay due to the injection process. Indeed, embryos cannot be injected too early when transcription hasn't started and as a consequence no fluorescence signal can be detected.

Is there a delay caused by the slow action of cycloheximide in the embryo?

As the embryos were injected prior to imaging, the observed dynamics arises from the combination of the incubation time that the drugs need to fully block translation and from the maturation time of already produced mNeonGreen molecules that were not yet fluorescing. These two processes are thus very difficult to distinguish from each other. Therefore, our estimate of the characteristic time taken by the fluorescence signal to reach a plateau in cycloheximide injected embryos, gives an upper bound for the maturation time of mNeonGreen. However, the value we obtained is in line with the estimate of mNeonGreen maturation time presented in another study (Lambert et al, Nat. Methods 2019). We are therefore confident that we can attribute the observed dynamics mainly to the maturation of mNeonGreen molecules.

Were these control experiments performed in triplicate?

As these experiments were controls, we didn't perform them in triplicates. To check the reproducibility of these injections, we injected and imaged two additional embryos for each drug, as shown below.

Cycloheximide Injections (translation inhibitor)

Amanitin Injections (transcription inhibitor)

As can be seen from the histograms, the maturation times and the mRNA degradation times obtained by the fit of the fluorescence time courses were well reproducible for all replicates. The different time delays and intensities of fluorescent signals come from slightly different stages of the embryos when they were injected.

In Sup Figures 3 and 4, the modeled mRNA production always deviates from the true production rate at late time points. Why is this happening? How does this error affect the analysis of the reporter measurements?

The reconstruction algorithm used in this study calculates the most likely timecourse of instantaneous and cumulative mRNA production using the information of the entire timecourse of fluorescence intensity. However, the mRNA production calculated at a specific time only depends on the fluorescence at later times. As a consequence, the mRNA reconstruction becomes less and less precise at later times, particularly when 10 or less data points are available.

Since the length of our measurements is limited by the onset of gastrulation, we decided to only present the result of the reconstruction up to $t=40$ or 45 min, which corresponds to 10 minutes before gastrulation, depending on the embryo.

Line 427: the labeling of the rate constant k_m and k_{dm} is inverted.

Line 524: Equation number is missing.

We thank the Reviewer for noting these typos that we corrected.

Reviewer #2 (Remarks to the Author):

Major Concerns:

This system, although shown to be excellent in a specific context, is not likely to draw a modestly broad readership. The manuscript focused on synthetic enhancer elements in very specific embryo context. Demonstration of a more broad utility for this would improve the manuscript. For example, how does it work with multiple non-synthetic enhancers and for enhancers active at additional developmental ages.

In the manuscript, we applied our mNeon green reporter with 4 different native and synthetic enhancers. Although we felt that this was sufficient to validate the new reporter system and demonstrate its applicability, we agree with the Reviewer that the paper would be improved in showing a broader application by measuring more enhancers. We thus expanded our study as follow:

We produced, imaged and analysed an additional embryo line carrying the native *kr_CD2* enhancer, which we believe substantially helps illustrating the potential of our method. The *kr_CD2* enhancer has a highly dynamic expression pattern consisting of two independent expression domains: an anterior domain established at an early stage of development, and a sharp central stripe, which is active later. Moreover, subtleties in the dynamic of expression of its central stripe domain have been recently investigated by two studies using the MS2-MCP reporter system. We included these results in the manuscript as a novel main figure (**Figure 3**). The data obtained with our reporter for the *kr_CD2* enhancer shows the well-defined expression of the two narrow stripes and their dynamics. These RNA patterns are perfectly consistent with what was reported using *in-situ* stainings (Segal et al. 2008) and the MS2-MCP system (El-Sherif & Levine, 2016; Scholes et al. 2019). Subtle effects like the dynamics shift of the central

stripe could also be observed in our data, in agreement with previous reports (El-Sherif & Levine, 2016). Moreover, the quantitative agreement of both the stripe position and width with published MS2-MCP data (Scholes et al. 2019) further confirms that our reporter can also capture sharp expression patterns.

We described **Figure 3** in the main text by adding on page 13 at line 211:

“To further illustrate the potential of our method to quantify the activity and dynamics of enhancers, we used the mNeon reporter to image an additional native *Drosophila* enhancer, *kr_CD2*, located in the Krüppel *cis*-regulatory region (Hoch et al., EMBO J. 1991)(**Figure 3**). The two native enhancers exhibit very different segmentation patterns and dynamics of total mRNA production (**Figure 3a-b**, average total mRNA for the three replicates of each enhancer). Whereas *hb_ant* shows its characteristic expression gradient along the AP axis (**Figure 3a**, higher and middle panels), *kr_CD2* contrasts with the well-defined expression of two strong, narrow stripes at ~20% and ~50% of the embryonic length, respectively. These RNA patterns are perfectly consistent with what was reported using *in-situ* stainings (Segal et al. 2008) and the MS2-MCP system (El-Sherif & Levine, 2016; Scholes et al. 2019). Previous works focused particularly on the central stripe of expression, and provided quantitative data on its dynamics using the MS2-MCP system (El-Sherif & Levine, 2016). These data revealed a dynamic shift of the stripe peak towards the embryo anterior, covering ~4% of the embryo length, in perfect agreement with the shift observed in our data (see inset in Fig. 3 d). The inspection of the temporal dynamics in the various domains reveals interesting features and differences between the *hb_ant* and *kr_CD2* *cis*-elements (**Figure 3a-c**, lower panels and Figure 3d). The total mRNA levels in the anterior (dots in **Figure 3d**) show similar dynamics with a gradual increase already before nc13 (our t=0). Interestingly, expression in the posterior stripes of *hb_ant* and *kr_CD2* starts later (after t=15mins; crosses in **Fig. 3d**), in agreement with the reported dynamics of the *kr_CD2* enhancer (El-Sherif & Levine, 2016). This differential time expression in the different regions may be related to the expression of later activators.

We then quantified the activity of synthetic enhancers. We tested two variants....”

We also want to mention that we are currently working on an additional manuscript in which we took advantage of this new method to measure the spatiotemporal dynamics of expression of 20 new synthetic enhancers. This additional study focuses on a specific biological question regarding the effect of binding sites for Hunchback in segmentation enhancers, which is beyond the scope of the current manuscript. We could however make this manuscript available for this Review.

Regarding the possibility of studying additional developmental stages, although this is in principle possible, it becomes substantially more difficult because of the rapid movements taking place in the embryo with the onset of gastrulation. To overcome this limitation one would have to track these rapid movements in order to measure the intensity of the mNeon signal in each cell over time. The high spatial and temporal resolution required to perform this analysis cannot be achieved with standard confocal microscopes, but could be achieved with other imaging techniques as, for example, light-sheet fluorescence microscopy.

We introduced a comment on this aspect in the main text of the manuscript on page 10 at line 166:

“In this work, we measure mRNA production dynamics focusing on stage 4 and 5 of embryo development. After this stage gastrulation occurs (stage 6) making more difficult the analysis due to the rapid movement of the nuclei (**Supplementary Fig. 7**).”

As a final remark, we want to stress that, although we optimized our approach to study enhancer activity in the very specific case of *Drosophila* embryo segmentation, the community that

study this system is large and very active. *Drosophila* segmentation is considered a perfect test system to study the mechanisms of enhancer activity. Moreover, for more than 20 years and still today, a number of studies focusing on this particular paradigm have been highly influential in shaping our understanding of the molecular mechanisms of transcriptional regulation. Therefore, we believe that our study will be an important contribution to a highly active and modestly broad community of researchers.

Minors Concerns:

Figure 1 panels c-e please annotate the identity of the enhancer driving reporter expression.

Figure 1, if EL% and AP% are the same thing, please pick one and stick to it.

Figure 2 has red squiggly line in label. Please remove.

Figure 2 legend. What is the arrow (it is stated in text, but not in the figure legend)? Change “data are pooled” to “data was pooled”

Line 97 and 107. The days of “data not shown” are over. Please add to supplement or else do not mention.

Line 140. Remove “/”

Line 195. Remove “a”

Figure 3a, scale bar is missing size in base pairs.

We thank the Reviewer for pointing out these errors and made the corrections.

Reviewer #3 (Remarks to the Author):

My comments are appended below:

1. What is the half-life of the reporter? For being a real-time reporter, fluorescent protein should be fast-degrading along with fast-maturing. It would be very informative if authors can show how long does it take the signal to turn off after the enhancer is silenced?

We didn't measure directly the lifetime of the protein, but assumed the protein degradation rate to be much lower than the mRNA degradation rate. This assumption was justified by the fact that, after blocking of translation, the fluorescence time course reaches a steady state for the remaining 30min (**Supplementary Fig. 2d** and the replicates shown above). This allowed us to estimate the degradation rates of mRNA (~35 mins) from the fluorescence time courses of the injection experiment (shown in **Supplementary Fig. 2f**, see **Supplementary Fig. 2g** and **METHODS** for details about the fitting procedure).

To check more substantially that the reporter has a long lifetime, we imaged *hb_ant* embryos over 5 hrs after nc14 (**Supplementary Fig. 7**). We focused on the fluorescence time course of the mNeon reporter for the anterior part of the embryo. Starting from stage 7 (corresponding to 50 mins after nc13, our time 0), the expression of the native hunchback locus is substantially shut down. Therefore, we can reasonably expect the expression driven by the specific enhancer examined in this study to also be shut down around the same time and that the synthesis of new mRNA molecules of the reporter stops. The observed fluorescence is compatible with this notion: translation of the mRNA molecules to mNeon fluorescent proteins results first in a strong increase of the fluorescence signal during stage 5-6 (highlighted in green in **Supplementary Fig. 7b**), followed by a more moderate increase during stage 7-9, which corresponds to translation of already produced mRNA that are being progressively degraded (highlighted in blue). Finally, once no mRNA molecules are present anymore the fluorescence signal exponentially decreases due to active protein degradation. As shown in **Supplementary Fig. 5b**, this decrease is not caused by photobleaching, but is solely due to active protein degradation. Hence, it is possi-

ble to extract from these data an estimate of the mNeon protein lifetime in this system by fitting the fluorescence time course after stage 10 with an exponential decay function. We found a characteristic protein lifetime of ~ 130 mins. This time is much longer than our inferred mRNA degradation lifetime of 35min, validating our assumption.

Note that in our system, the fluorescence protein should be fast-maturing as the Reviewer mentioned, but not fast-degrading as we extract mRNA levels from the time evolution of the fluorescence signal rather than from its absolute values (see **Fig. 1b, f-i** and **Supplementary Fig. 1**).

2. Further, a PEST domain can be added to the reporter to reduce its half-life. Although, this is required only if the half-life of the reporter is very long and signals are seen even if the enhancer activity is turned-off.

A PEST domain is indeed usually used to increase time sensitivity in standard fluorescence reporter assays. However, reducing the reporter lifetime results in lower signal, impeding detection sensitivity. Hence, there is always a tradeoff between time resolution and detection sensitivities. One of the advantages of our mNeon based reporter system is precisely not to need any additional sequences that would reduce protein lifetime. Indeed, we detect mRNA production not directly from the reporter fluorescence, but from its dynamical changes over time. This procedure is compatible with a long lifetime of the reporter, which is pivotal in providing a high sensitivity.

3. The authors have used the hb_ant enhancer to show that the mNeonGreen signal is localized along with the spatiotemporal domain of hb_ant. The expression domain of hb_ant is very wide, so to confirm spatial restriction of the reporter it will be better to use enhancers having a smaller expression domain like pair-rule enhancer or any segment-polarity enhancer that expresses in segments.

We agree with the Reviewer that the paper would benefit from presenting additional enhancers. As mentioned above, we included in this revision one additional native enhancers: *kr_CD2*, which has a smaller, striped expression domain (**Figure 3**). For the central stripe of expression of *kr_CD2* in particular we measure sharp striped pattern, in perfect agreement with *in situ* hybridization staining images.

4. Authors have provided conclusive evidence that the reporter can be used to check the activity of weak enhancers like Bicoid which proves that the reporter is very sensitive.

5. Also, authors have shown that adding Zld binding sites can increase the activity of weak enhancers and hence has the potential to facilitate the study of weak enhancers.

We thank the Reviewer for these positive comments.

Reviewers' comments:

Reviewer #1 (Remarks to the Author):

The authors have addressed all my comments. I recommend the publication of this paper.

Reviewer #2 (Remarks to the Author):

The authors made satisfactory major revisions to their manuscript, and satisfactorily addressed all of my previous comments. Additionally, the authors responses to the comments made by the other two reviewers were satisfying to me. Thus, I recommend this revised manuscript for publication.

Reviewer #3 (Remarks to the Author):

I agree with the general outcome of the paper that the rapidly maturing fluorescent reporters used by authors provide a more detection sensitivity for weak promoters. The revised manuscript addresses some of the major concerns of the half-life of the reporter, spatial confinement of the reporter (or leakiness) by using *kr_CD2* enhancers. The advantage of this system is definitely its detection sensitivity that can help to detect signals from weak promoters as well. Also, because of the spatial confinement of the signal, it can be used to study the promoters with narrow range expression patterns.

However in this system, the time resolution issue cannot be resolved as it will decrease the detection sensitivity. Therefore, this system can only be used to study when the promoter is turned ON during development, but it cannot be used to study how long the promoter is turned ON, or at what time is the promoter turned-OFF.

Response to reviewers

Reviewers' comments:

Reviewer #1 (Remarks to the Author):

The authors have addressed all my comments. I recommend the publication of this paper.

Reviewer #2 (Remarks to the Author):

The authors made satisfactory major revisions to their manuscript, and satisfactorily addressed all of my previous comments. Additionally, the authors responses to the comments made by the other two reviewers were satisfying to me. Thus, I recommend this revised manuscript for publication.

Reviewer #3 (Remarks to the Author):

I agree with the general outcome of the paper that the rapidly maturing fluorescent reporters used by authors provide a more detection sensitivity for weak promoters. The revised manuscript addresses some of the major concerns of the half-life of the reporter, spatial confinement of the reporter (or leakiness) by using `kr_CD2` enhancers. The advantage of this system is definitely its detection sensitivity that can help to detect signals from weak promoters as well. Also, because of the spatial confinement of the signal, it can be used to study the promoters with narrow range expression patterns.

However in this system, the time resolution issue cannot be resolved as it will decrease the detection sensitivity. Therefore, this system can only be used to study when the promoter is turned ON during development, but it cannot be used to study how long the promoter is turned ON, or at what time is the promoter turned-OFF.

We thank the Reviewer for the opportunity to clarify this very important point of our work. Within the limits imposed by the time resolution of our method, which we calculate to be ~ 7 minutes, our approach is able to determine both when an enhancer is turning on and when it is turning off.

In order to directly demonstrate that the algorithm is able to detect these transitions, we extended the set of simulations and present the results in a new supplementary figure (Supplementary Fig. 4), which is referenced in the main text (page 8, line 141). For these simulations we considered three basic scenarios, in which the enhancer is either already active at the beginning of the measurement, only active in a limited time window, or is activated at some point during the measurement and remains active until the end of the experiment. In all these cases, the algorithm was able to accurately detect the dynamics, both at the level of instantaneous mRNA production rate and for cumulative mRNA production.

The ability of our method to capture both of these aspects of an enhancer's dynamics can also be appreciated in the behaviour of the enhancers studied in this work. For example, in the case of the *hb_ant* enhancer, one can see how the total (or cumulative) mRNA production that we present in Fig. 1 h clearly saturates for many traces at $t \sim 20$ mins: this is the mark of the enhancer turning off in those portions of the embryo at the same time. This is probably even clearer when looking at the instantaneous rate of mRNA production (Supplementary Fig. 7 c) which rapidly decreases after $t \sim 20$ mins. To make this point more explicit we included in the revised version of the manuscript a comment at line 170, page 10:

“During the second half of nc 14, the enhancer is gradually turned off, as can be seen from the saturation of cumulative mRNA levels (**Fig. 1h**).”

Enhancer activity can be represented equivalently by the total mRNA production or by the instantaneous rate of mRNA production. In this work, we decided to present our data mainly in terms of total mRNA production because we are primarily interested in the spatial domains of enhancer activity, for which the cumulative levels offer a clearer and less noisy representation. Similarly, previous studies based on the MS2-MCP system have also employed cumulative mRNA levels to represent the activity domain of an enhancer (e.g. Garcia et al. 2013). However, for all the enhancers studied in this work, we do provide both the instantaneous mRNA production rate and the total mRNA levels for each biological replicate, as part of the supplementary materials.

REVIEWERS' COMMENTS:

Reviewer #1 (Remarks to the Author):

The authors had already addressed my comments in the revised version of the manuscript. In this new version, they provide further evidence that they are able to use the fluorescence intensity dynamics to recover the transcriptional induction kinetics. I recommend the publication of this manuscript.

Reviewer #3 (Remarks to the Author):

The authors have addressed all my comments. I recommend the publication of this paper.